# Effects of Postnatal Caffeine Exposure on Absence Epilepsy and Comorbid Depression: Results of a Study in WAG/Rij Rats

**DOI:** 10.3390/brainsci12030361

**Published:** 2022-03-08

**Authors:** Gul Ilbay, Zeynep Ikbal Dogan, Aymen Balıkcı, Seyda Erdogan, Akfer Karaoglan Kahilogulları

**Affiliations:** 1Department of Physiology, Faculty of Medicine, Kocaeli University, Kocaeli 41100, Turkey; zynp_ikbal@hotmail.com (Z.I.D.); pt_eymen@hotmail.com (A.B.); 2Department of Neurology, Faculty of Medicine, Ankara University, Ankara 06560, Turkey; dr_seyda@yahoo.com; 3Department of Psychiatry, Health Sciences University Dışkapı Yıldırım Beyazıt Training and Research Hospital, Ankara 06500, Turkey; akferkaraoglan@hotmail.com

**Keywords:** chronic absence seizure, caffeine, depression, WAG/Rij rats

## Abstract

The present study aims to investigate effect of early caffeine exposure on epileptogenesis and occurrence of absence seizures and comorbid depression in adulthood. For this purpose, Wistar Albino Glaxo from Rijswijk (WAG/Rij) rats were enrolled in a control and two experimental groups on the 7th day after the delivery. The rats in experimental groups received either 10 or 20 mg/kg caffeine subcutaneously while animals in control group had subcutaneous injections of 0.9% saline. The injections started at postnatal day 7 (PND7) and were continued each day for 5 days. At 6–7 months of age, electroencephalogram (EEG) recordings and behavioral recordings in the forced swimming test, sucrose consumption/preference test and locomotor activity test were carried out. At 6 months of age, 10 mg/kg and 20 mg/kg caffeine-treated WAG/Rij rats showed increased immobility latency and active swimming duration in forced swimming test when compared with the untreated controls. In addition, 20 mg/kg caffeine treatment decreased immobility time. In sucrose preference/consumption tests, WAG/Rij rats in 10 mg/kg caffeine group demonstrated higher sucrose consumption and preference compared to untreated controls. The rats treated with 20 mg/kg caffeine showed higher sucrose preference compared to control rats. The exploratory activity of rats in the 10 mg/kg caffeine-treated group was found to be higher than in the 20 mg/kg caffeine-treated and control groups in the locomotor activity test. At 7 months of age, caffeine-treated animals showed a decreased spike-wave discharge (SWD) number compared to the control animals. These results indicate that postnatal caffeine treatment may decrease the number of seizure and depression-like behaviors in WAG/Rij rats in later life. Caffeine blockade of adenosine receptors during the early developmental period may have beneficial effects in reducing seizure frequency and depression-like behaviors in WAG/Rij rat model.

## 1. Introduction

Depression is a common mental disorder affecting 280 million people worldwide [1]. Despite the availability of effective treatments for mental disorders including depression, people who experience depression are often not diagnosed correctly or receive no treatment [2]. Absence seizure incidence varies from 0.7 to 4.6 per 100,000 in the general population [3]. Similar to depression, many people with epilepsy do not receive appropriate and adequate treatment for their condition. This is defined as a phenomenon called the treatment gap.

Many developing countries with limited resources struggle to develop and implement policies to prevent an increase in the treatment gap. Prevention or early diagnosis and intervention programs have been developed to reduce the burden of these two disorders. Effective community-based programs have been shown to reduce depression [1]. However, development of psychosocial interventions is not enough to prevent all depression cases due to the multifactorial nature of the disorder including biological, environmental and personal factors. Moreover, there is still a group of patients who do not respond to current treatments even when they access quality health care in a timely fashion.

Epileptogenesis is defined as the development and extension of tissue which is able to generate spontaneous seizures and refers to the process by which the previously normal brain is functionally altered towards the generation of abnormal electrical activity [4,5]. Epileptogenesis is considered to occur in three phases: occurrence of a precipitating injury or event, a latent period and chronic, established epilepsy. The latent period of epileptogenesis is a clinically silent period characterized by ongoing remodeling processes that may last from days to years after the injury. Several mechanisms may be responsible for epileptogenesis, including plasticity and trafficking of GABA-A receptors, network re-organization, increased inflammatory processes, changes in gene and receptor expression and seizure or status epilepticus-induced alterations in ion channels. Recent data indicate that epileptogenesis is not restricted over a limited time—contrary to what was previously thought—and continues to progress after the expression of spontaneous seizures making the process a moving target [5,6]. Therefore, interventions during this period might be used both to prevent the subsequent development of epilepsy and disease modification. From a therapeutic perspective, disease modification consists of antiepileptogenesis and comorbidity modification. Antiepileptogenic treatment can be given before or after the epilepsy onset and may result in complete prevention or delay of epilepsy development or reduce the severity of the disease. The other step of treatment is preventing the burden of comorbid diseases such as anxiety, depression, somato-motor impairment or cognitive decline. Despite the large number of anti-seizure medications, they fail to prevent epilepsy development and provide disease modification. Even seizure control cannot be achieved in up to 30% of epilepsy patients with current anti-seizure drugs. Thus, the development of novel therapeutic approaches to prevent and treat epileptogenesis and its comorbid disorders is crucial. There have been an increasing number of studies investigating the antiepileptogenic effects of different medications in animal models of structural and genetic epilepsies.

A number of clinical observations show a relationship between epilepsy and depression. Both depressive disorders and epilepsy have a genetic basis. In 1986, researchers described one of the most well-known rat models as a valid genetic animal model of absence epilepsy with comorbidity of depression absence epilepsy [7]. Rats of Wistar Albino Glaxo from Rijswijk (WAG/Rij) strain, around 2 months of age, showed spontaneous, generalized spike-wave discharges (SWDs) in electroencephalograms (EEG) along with decrease of consciousness and an interruption of continuous activity, as in human absence epilepsy. After the age of six months, around 16–20 SWDs/h appeared in the EEG of WAG/Rij rats and continued in a lifelong manner, therefore serving as the chronic animal model for absence epilepsy [8]. WAG/Rij rats are also confirmed as an animal model for chronic depression since they exhibit depression-like symptoms [9]. Depression-like behavioral symptoms emerge at 3–4 months of age and tend to be aggravated parallel to the increase of SWDs [10].

The molecules investigated in animal models of genetic epilepsies include anti-seizure drugs such as levetiracetam, ethosuximide, zonisamide, vigabatrin and perampanel; antidepressants such as clomipramine, fluoxetine and duloxetine; statins such as atorvastatin, simvastatin and pravastatin, immunosuppressants or immunomodulator drugs such as rapamycin, etericobix and fingolimod [5,6,11]. Early long-term fingolimod treatment was shown to have antiepileptogenic and antidepressant effects in WAG/Rij rats, however the antiepileptogenic effects were transitory given that the absence seizures returned to control levels 5 months after treatment discontinuation [11]. The effect of liraglutide—a novel glucagon-like peptide-1 analogue—on epileptogenesis and comorbid behavioral alterations was investigated in the mouse intrahippocampal kainic acid model of temporal lobe epilepsy and the WAG/Rij rat model of absence epilepsy [12]. It was proposed that liraglutide had neuroprotective effects with several mechanisms including reduced neuroinflammation and increased pro-survival cell signaling. Liraglutide showed antiepileptogenic effects and prevented comorbid memory impairment and anxiety-like behavior in kainite induced epilepsy. However, there was no modifying effect on epileptogenic process in the WAG/Rij rat model of absence epilepsy. Recently, the disease modifying effect of two histone deacetylase inhibitors, sodium butyrate and valproic acid and their co-administration, was studied in WAG/Rij rats, based on the evidence pointing to the link between epigenetic mechanisms such as alterations in histone acetylation and the epileptogenic process [13]. The results of the study demonstrated the antiepileptogenic effects of each drug with enhancement by co-administration. Additionally, one month after treatment withdrawal, depressive-like behavior and cognitive performance showed improvement with early-chronic treatment. Therefore, the investigators proposed histone deacetylase inhibitors as a potential candidate in management of epileptogenesis and psychiatric comorbidity.

The evidence of environmental factors leading to epigenetic modifications in the development of epilepsy and comorbid depression have also been shown in the WAG/Rij rat model [14]. The onset of absence epilepsy was delayed, and depression symptoms were reduced in the WAG/Rij pups that received intensive early maternal care by their foster Wistar mothers [15]. Our previous study showed that early neonatal tactile stimulations may decrease seizure activity and comorbid depression-like behaviors at adult ages in WAG/Rij rats [16]. These examples demonstrate the effect of early environmental factors on genetic absence epilepsy and comorbid depression.

Environmental factors, modulating adenosine, have recently been reported as other factors that can control the development of epilepsy [17]. Adenosine, a neuromodulator that terminates seizures in the brain through activation of adenosine A_1_ receptors and decreases neuronal excitability also regulates cognitive and psychiatric behavior [18,19]. Gomes et al. reported that major depression significantly affects the expression and functioning of the adenosinergic system, with data on A1R and A2AR [20].

Due to having molecular structural similarities to adenosine, caffeine can bind to the A_1_ and A_2A_ adenosine receptor subtypes and block adenosine from binding thereby acting as an adenosine antagonist [21].

There are numerous studies investigating chronic low-dose caffeine effects on seizures and epilepsy in adult rats and they demonstrate different outcomes. Some of these studies did not find a relation between exposure to chronic low-dose caffeine or withdrawal from it and seizures [22]. However, there are studies reporting decrease in the duration of convulsions in animals that received caffeine for several days in comparison to ones that received saline, in addition to studies reporting seizures triggered by a single-high caffeine dose [23]. In adult rats with genetic absence epilepsy, seizure activity decreased after an acute dose of caffeine, while chronic doses had no effect [24]. Moreover, seizure susceptibility was demonstrated to decrease in young animals by chronic, low-dose caffeine exposure. Rat pups exposed to chronic low doses of caffeine in the first weeks after birth showed an increase in seizure threshold for generalized tonic-clonic seizures in adulthood [13]. Tchekalarova et al. demonstrated that neonatal caffeine treatment caused suppression of rhythmic metrazol activity evoked by PTZ in adult rats in another study [25]. Overall, the impact of caffeine on seizure seems to be dependent on age, seizure model, caffeine dose, and administration method [23].

Studies on caffeine and its relationship to depression are not as numerous as the ones on epilepsy but there are promising results about the impact of caffeine as an adenosine receptor antagonist on symptoms of depression. Epidemiological studies report beneficial effects of intermediate levels of caffeine consumption (300–550 mg/day) on depressive symptoms in non-clinical populations while higher doses have negative effects [26]. These effects are also shown in animal models [27].

Considering the burden of the disorders and high comorbidity, we have investigated the effects of early intervention of caffeine on comorbid epilepsy and depression in the chronic animal model of absence epilepsy and comorbid depression.

## 2. Materials and Methods

### 2.1. Ethical Approval

Data from the experimental study that was approved by the Ethics Committee of Kocaeli University (KOU-7/1-2016) is revisited for this paper.

### 2.2. Animals and Caffeine Treatment

Progeny of pregnant WAG/Rij rats kept in separate cages in an environment at 22–23 °C and on a 12–12 h light–dark cycle with free access to water and food were enrolled in the study. The births were monitored and on the 7th day following birth (PND7), pups were randomly assigned to either a control group or to one of the two experimental groups. Rat pups in experimental groups were given subcutaneous injections of 10 or 20 mg/kg caffeine. The control group were injected with 0.9% saline. The injections started at PND7 were continued for 5 days. We chose the period of 7–11 days for the intervention since adenosine receptors are present but not fully mature in the various brain areas at this developmental stage [19]. Rat pups stayed with their mother and were separated just for a short amount of time during injection. Litters were weaned on postnatal day 21. After weaning, rats were housed in different cages according to the experimental groups and were maintained under normal conditions until 180 days of age. 

### 2.3. Behavioral Tests

All WAG/Rij rats (*n* = 9 for saline, *n* = 10 for 10 mg/kg caffeine and *n* = 9 for 20 mg/kg caffeine treatment) were individually exposed to a forced swimming test (FST), sucrose consumption test (SCT) and locomotor activity test at the age of 6 months between 10:00 am and 3:00 pm each day of the tests. 

#### 2.3.1. Force Swimming Test

Depressive-like behaviors are assessed with FST, which has been used in many studies with some minor modifications [14,28]. The experiment was conducted in a transparent cylinder (47 × 38 cm) filled with water 22 ± 1 °C. Initially, rats (PND180) were forced to swim for 15 min as part of the pretest session and were dried before being put back into their cages. Twenty-four hours after the pretest session, rats were resubmitted to the forced swimming test for 5 min and their swimming behavior was recorded using video cameras. Immobility (duration of passive swimming), immobility latency and swimming time (total duration of active swimming) were assessed by a blinded rater for each rat. Immobility is defined as having no additional activity other than the required act of keeping the head above water.

#### 2.3.2. Sucrose Consumption/Preference Test

Anhedonia and motivation were assessed with the sucrose consumption/preference test. The drinking of sucrose solution (20%) and the number of approaches to the bottle were calculated during fifteen minutes for each rat. The bottles were weighed at the beginning and at the end of the test in order to measure the sucrose intake. Animals were not subjected to any of food and water deprivation. Values of sucrose consumption on the 4th day were used to compare the differences between rat groups after a 3-day adaptation procedure as defined in previous studies [29].

#### 2.3.3. Locomotor Activity Test

A locomotor activity was recorded by the rat activity monitoring system (Commat Ltd., Ankara, Turkey) consisting of a Plexiglas test chamber, computer and activity software. Rats were taken to the test room for 1 h and were then placed into plexiglass chamber (42 × 42 × 30 cm) of the system. The chamber had infrared photocells, with pairs of 15 infrared photobeams and detectors located every 2.5 cm in the horizontal plane (bottom) and every 4.5 cm in the vertical plane (upper). Locomotor activities measured with interruption of photocell beams were recorded by computer for 10 min. The repeated beam breaks at the same photobeams are used as a measure of stereotypic activity. The breaks in the upper set of photobeams were used as a measure of vertical activity. Breaking more than 1 consecutive photobeam in the bottom set of photobeams was used as a measure of ambulatory activity. Total distance travelled (in cm) and numbers of stereotypic, ambulatory and vertical movements were analyzed [16].

### 2.4. Surgery, EEG Recordings and Assessment of Absence Seizures

At 7 months of age, 7 animals in each group were equipped with electrodes. Tripolar electrodes (MS3333/2A; Plastic One, the United States) were put over the frontal (AP 2.0 mm, L 3.5 mm), the parietal (AP −6.0 mm L 4.0 mm) and the cerebellar cortex (reference electrode) with stereotaxic surgery under Xylazin (5 mg/kg ip) and Ketamine (60 mg/kg ip) anesthesia. After 2 weeks of recovery, animals were connected to the MP150 EEG recording system. After the rats that were free to move were habituated to recording conditions for 1 h, EEG recordings from each subject were taken for 4 h between 10:00 am–2:00 pm. Numbers and durations of spike wave discharges (SWD) which met the following criteria defined by van Luijtelaar et al. were analyzed: duration 1–10 s, having spike and wave with frequencies between 7–10 Hz and doubled amplitude to background activity [30].

### 2.5. Statistical Analyses

Data are as means ± S.E.M. The analyses were run in GraphPad Prism 7.03 (San Diego, CA, USA), using a one-way ANOVA followed by post hoc tests. Figure legends demonstrate statistical tests and sample sizes. The level of 0.05 was made use of as a threshold for statistical significance.

## 3. Results

### 3.1. Behavioral Measures in the Forced Swimming Test

In rat pups treated with caffeine, latency to immobility was significantly longer than the rats in the control group (10 mg/kg vs. control *p* = 0.023; 20 mg/kg vs. control *p* < 0.001) (Figure 1A). The immobility durations of the rats in the group treated with 20 mg/kg caffeine injection were significantly less than for rats in the control group (*p* < 0.05) (Figure 1B). The 10 mg/kg caffeine group also showed shorter immobility duration compared with the control group (Table 1) but the difference was not statistically significant. 

When swimming time was evaluated, it was found that the rats in the 10 mg/kg and 20 mg/kg caffeine treatment groups were significantly more active than the rats in the control group (10 mg/kg vs. control *p* < 0.05; 20 mg/kg vs. control *p* < 0.01) (Figure 1C).

### 3.2. Sucrose Consumption/Preference Test

Sucrose consumption in the 10 mg/kg caffeine treatment group was significantly higher compared to the control group (*p* = 0.01) (Figure 2A). Caffeine treatment groups had a higher sucrose preference (%) compared to the control group (10 mg/kg vs. control *p* < 0.001; 20 mg/kg vs. control *p* < 0.05) (Figure 2B).

### 3.3. Locomotor Activity

In the locomotor activity test, significant differences amongst the 10 mg/kg caffeine treatment group, the 20 mg/kg caffeine treatment group and the control group were found. The locomotor activity distance of the 10 mg/kg caffeine-treated group was significantly higher in comparison to the control group (*p* < 0.01) (Figure 3A). Mean number of ambulatory activities in the 10 mg/kg caffeine treatment group was significantly higher than the other two groups (10 mg/kg vs. control *p* = 0.001; 10 mg/kg vs. 20 mg/kg caffeine *p* < 0.05) (Figure 3B). Rats treated with 20 mg/kg caffeine showed significantly higher stereotypic movements than those of the 10 mg/kg caffeine treatment and the control group (*p* = 0.01) (Figure 3C). No change was observed amongst the groups in vertical movements. 

### 3.4. Effects of Postnatal Caffeine Treatment on SWDs in Adult WAG/Rij Rats

SWDs were detected on the EEG background, which are repetitive complexes with a frequency of 7–10 Hz (Figure 4). SWD number and total duration were summed for 4 h in caffeine-exposed and control WAG/Rij rats.

Figure 5 depicts the results for number and total duration of SWDs in three groups. Rats in groups treated with caffeine showed less SWD number; however, a statistically significant decrease was found only in the 20 mg/kg caffeine treatment group (*p* < 0.05) (Figure 5). When total SWD duration was compared, no statistically significant difference was found between the groups but total durations of SWD of the 10 and 20 mg/kg caffeine-treated group were lower than control group. Total SWD duration was 850 ± 150 in the control group, 489 ± 114 in the 10 mg/kg caffeine-treated group and 656 ± 154 in the 20 mg/kg caffeine-treated group.

## 4. Discussion

Caffeine is a central nervous system stimulant, and several interactions may contribute to its stimulative effects. For instance, it modulates the GABA-A receptors and changes the response to inhibitory gamma-aminobutyric acid. Additionally, the molecular structure of caffeine (1,3,7-trimethylxanthine) is similar to adenosine [31]. Adenosine is a purine and two different types of purinergic receptors have been described: selective for adenosine (P1) and selective for ATP/ADP (P2) [32]. The adenosine selective P1 receptors, have four subtypes with A1 and A2A subtypes showing high affinity for adenosine. Action of A1 receptors results in inhibition of adenylate cyclase followed by inhibition of glutamatergic transmission. The effect of seizure suppression of adenosine, as an inhibitory neuromodulator is supposed to be mainly based on A1 receptors [33]. Caffeine is a competitive non-selective P1 receptor antagonist binding to the A1 receptor as well as the A2A receptor [23]. Contrary to A1 receptors, the action of A2A receptors results in activation of adenylate cyclase further causing an increase in glutamatergic transmission [34]. Accordingly, the answer to the question of the effects of caffeine on the development of seizure activity is not simple.

Previous research showed that caffeine may cause neuronal hyperexcitability by increasing Ca^2^ influx, changing potassium currents and antagonism of inhibitory A1 adenosine receptors [23,34,35]. In fact, caffeine was previously proposed as an animal model of seizures [36]. However, the results of animal studies have suggested the effect may be towards either increasing seizure susceptibility or protecting against seizures; depending on age, the animal epilepsy model, administration type (a single dose or long-term exposure) and caffeine dose [23]. The results of studies investigating the effects of caffeine exposure in adult animals are conflicting. Interestingly, a recent study investigating the effect of agonists and antagonists of adenosinergic activity on seizures and inflammation in WAG/Rij rats found that caffeine administered rats show a decreased SWD number on EEG linked to an increase of cytokine levels in the thalamus [37]. The results of this study may be important for pointing to the potential interactions between the adenosinergic system, inflammation and seizures. Indeed, adenosine A2A receptors are found in immune cells and involved in tissue protection by reducing inflammation [38]. The role of adenosine A2A and P2 receptors via inflammatory processes attracts attention in other neurological disorders such as Parkinson’s disease, memory impairment, multiple sclerosis or amyotrophic lateral sclerosis [34].

Animal studies on the effect of maternal caffeine exposure reported an increased seizure susceptibility by hyperthermia in the offspring of rat dams receiving caffeine during pregnancy [39]. Additionally, a delayed migration of GABAergic neurons into the hippocampus was reported. In humans a few studies pointed to this issue. A study analyzing 35,596 children in Denmark found no association between maternal caffeine intake and the risk of febrile seizures in the first three months of life [40]. So far, in humans there is no clear evidence that maternal caffeine intake increases the risk of seizures.

The protective effects of caffeine stand out especially when administered to young animals having an immature brain, as in our study [23]. This may be most likely due to changes in the density and sensitivity of adenosine A1 and A2A receptors during brain development at young ages [41,42,43]. The present study showed that repeated treatment with caffeine from postnatal day 7 to 11 has an impact on EEG parameters of absence seizures and depression-like behaviors in adult WAG/Rij rats that have a genetic predisposition to absence epilepsy and comorbid depression. Caffeine treatment groups have a decrease in both number and total duration of SWDs. Caffeine treatment groups showed increased immobility latency, swimming time and reduced immobility duration in the forced swimming test. Sucrose preference and sucrose consumption were also increased with caffeine treatment. Caffeine-treated rats were more active and showed more stereotypic and exploratory activity in the locomotor activity test in comparison to the control rats.

There are experimental studies which suggest that seizure susceptibility could be modified by early caffeine treatment during development. The findings of these studies indicated that early caffeine exposure exerts predominantly anticonvulsant effects, but the effects may change by age, dose and model [44]. For example, caffeine treatment during the second postnatal week (P7–P11) resulted in decreased seizure threshold to PTZ and picrotoxin while they showed full resistance to bicuculline at 25-day-old Wistar rats [45]. Data on the effect of early caffeine treatment on seizure activity in adulthood is limited. Postnatal caffeine treatment was shown to decrease the seizure susceptibility to different convulsive agents such as PTZ, picrotoxin and kainic acid in adult rats [46]. Tchekalarova et al. showed that a chronic low dose of caffeine 7–11 days after birth caused suppression of seizures in a PTZ model absence seizure [13]. Similar to these convulsive model results, postnatal caffeine treatment induced suppression on seizure expression in adulthood in our nonconvulsive model of genetic absence epilepsy. The results of our study support the idea that caffeine exposure during the developmental period can have a permanent effect on seizure activity in adulthood. Common suggested mechanisms explaining the effect of caffeine on seizure activity are its inhibition on adenosine receptors and permanent changes of adenosine receptor subtypes in different brain areas [47]. Effects of neonatal caffeine on number of SWDs could be associated with blockade of the A_1_ and A_2_ receptor subtypes in our study; Adenosine receptors exist but are immature in the second neonatal week (P7–P11) of development that we administered caffeine [44].

It was previously pointed out that the adenosinergic system might have been involved in modulation of absence epilepsy and depression mechanisms.

A firm, well-documented relationship is defined between SWDs and vigilance level. Absence seizures occur more during drowsiness and light slow-wave sleep and less often in high arousal states [48]. Considering this relation with occurrence of SWDs and vigilance level, the reduced seizure activity could be associated with increased locomotor activity in animals treated with caffeine. Enhanced locomotor activity was also previously reported in rats treated with caffeine at postnatal days 7–11 [49].

It can also be suggested that caffeine exposure may induce epigenetic modifications, and these epigenetic changes can lead to long-lasting effects in brain excitability by affecting the adenosinergic system. Pathological changes in DNA methylation homeostasis, which underlie epileptogenesis, were previously demonstrated [50]. It is known that a number of genes are involved in the development of absence epilepsy in WAG/Rij rats. The epileptic phenotype (SWDs) in WAG/Rij rats is determined by one gene with a dominant mode of inheritance while other genes determine the number of seizures. The number and duration of SWDs are determined by different genes [14]. Treatment with caffeine reduced the number of SWDs significantly while the decrease in the total duration of SWDs was not statistically significant. It seems like early-life caffeine exposure has influence on mechanisms of seizure generation rather than seizure termination. It should be emphasized that SWD occurrence is triggered in the somatosensory cortex while reticular thalamic nucleus is known to be associated with SWD termination [51].

In this study, we found that postnatal caffeine treatment has a positive impact on depression as measured with forced swimming and sucrose preference tests. These are the first data on the antidepressant-like effects of postnatal caffeine exposure. It has been known that caffeine leads to alteration in mood and anxiety in humans and animals [27]. Caffeine in low doses induces antidepressant effects without any major negative consequences on health [52]. Experimental studies also suggested that the adenosinergic system and its nonselective receptor antagonist caffeine have an important role in the regulation of depressive-like behavior [53]. Caffeine and selective A_2A_ receptor antagonists reduce immobility in the forced swimming test and the tail suspension test in mice which shows their potential role as antidepressants [54,55]. Chronic caffeine exposure was also found to reduce anxiety and depression in adult animals in chronic unpredictable stress models [52].

On the other hand, the relationship between depression and presence of SWDs was shown in genetic absence epilepsy models [9]. Depression-like behaviors are triggered by occurrence and repetition of seizure activity. Therefore, reduced depression-like symptoms in the caffeine treated rats might be the consequence of reduced seizure activity.

We used forced swimming and sucrose preference tests to measure depression-like symptoms in this study. A forced swimming test is generally used in evaluating antidepressant effect. The test in general has numerous advantages but it may be influenced by factors such as sleep abnormalities, or it may reflect an adaptive behavior to survive. We did not evaluate sleep abnormalities, which is a limitation for the study. Considering the methodology used, we do not think the differences in the forced swimming test are linked to development of an adaptive behavior. In this study, results of the forced swimming test and sucrose preference test are not consistent for different doses of caffeine administration. The inconsistency between two tests may be related with the symptoms that the tests aim to measure; it has been assumed that the sucrose preference test measures “anhedonia” while the forced swimming test measures “despair”. Unfortunately, we cannot explain the reason behind the lack of dose-dependency for the sucrose preference test shown in this study.

In summary, the present study results show that postnatal intervention of caffeine reduces the frequency of absence epileptic seizure and comorbid depressive symptoms and leads to permanent locomotor hyperactivity in adulthood. It can be suggested that the blockade of adenosine receptors by caffeine during the early developmental period mediates the effects of caffeine on absence epilepsy, depressive-like behaviors and locomotor activity in adulthood in WAG/Rij rats. Further research is required to explore the role of caffeine as an epigenetic regulator on the prevention of epileptogenesis and depressive-like symptoms in absence epilepsy.

## Figures and Tables

**Figure 1 brainsci-12-00361-f001:**
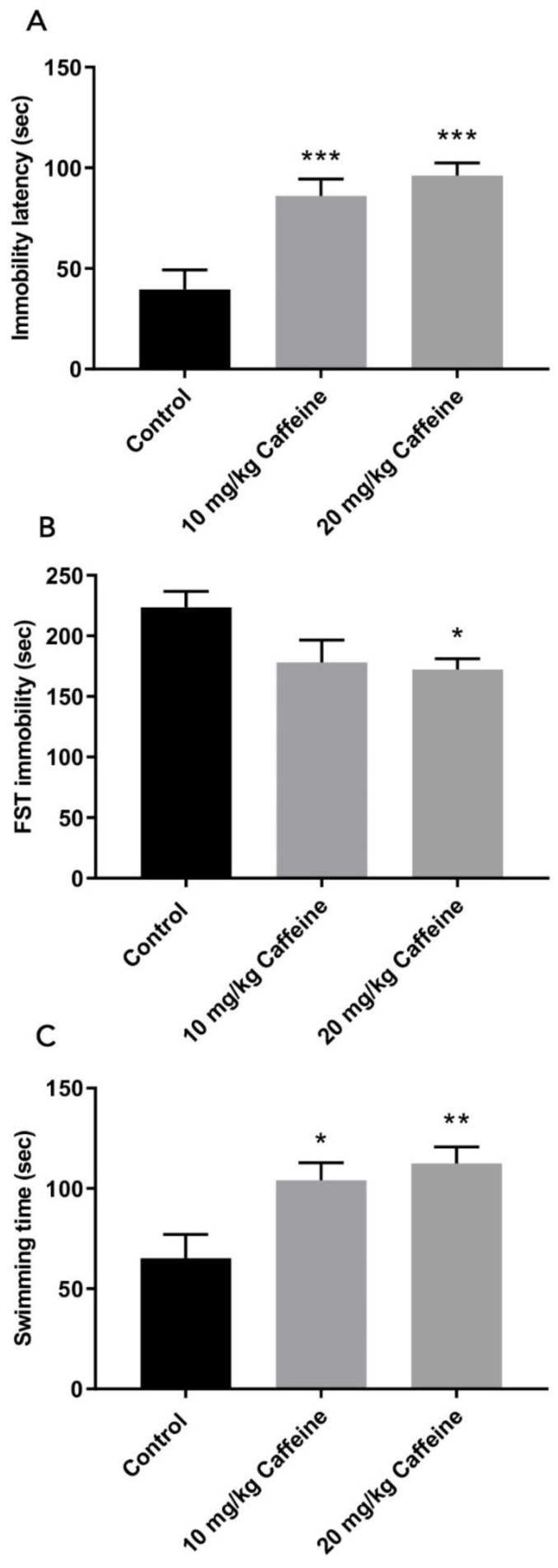
Behavioral measures in the forced swimming test control (*n* = 9), 10 mg/kg (*n* = 10) and 20 mg/kg (*n* = 9) caffeine-treated WAG/Rij rats. Data are shown as mean ± SEM. Changes in (**A**) Latency until first immobility period, (**B**) Time spent in immobility, (**C**) Swimming time. Data marked with asterisks are significantly different. * *p* < 0.05, ** *p* < 0,01 *** *p* ≤ 0.001 versus the control group (one-way ANOVA with post hoc Tukey test).

**Figure 2 brainsci-12-00361-f002:**
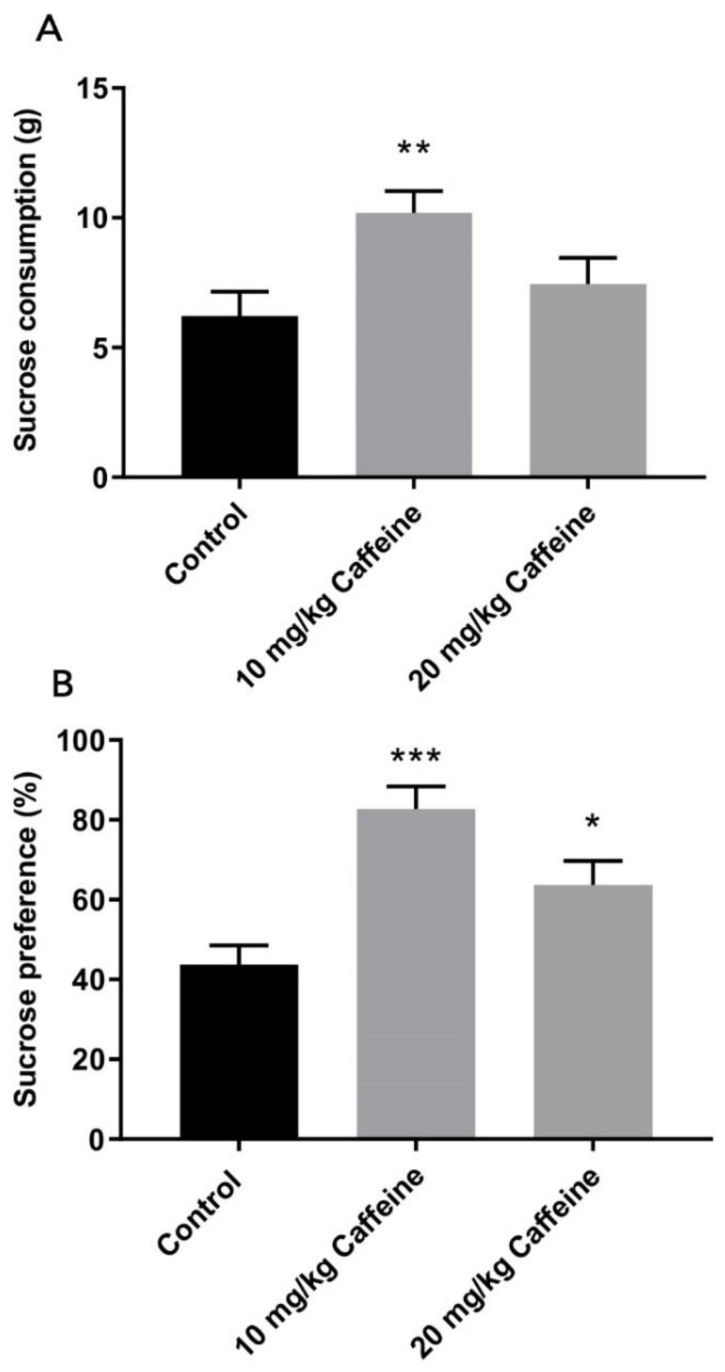
Effects of postnatal caffeine treatment on sucrose consumption (**A**) and preference (**B**) in control (*n* = 9), 10 mg/kg caffeine-treated (*n* = 10) and 20 mg/kg caffeine treated (*n* = 9) WAG/Rij rats. Data are shown as mean ± SEM. Data marked with asterisks are significantly different. * *p* < 0.05, ** *p* ≤ 0.01, *** *p* < 0.001 versus the control group (one-way ANOVA with post hoc Tukey test).

**Figure 3 brainsci-12-00361-f003:**
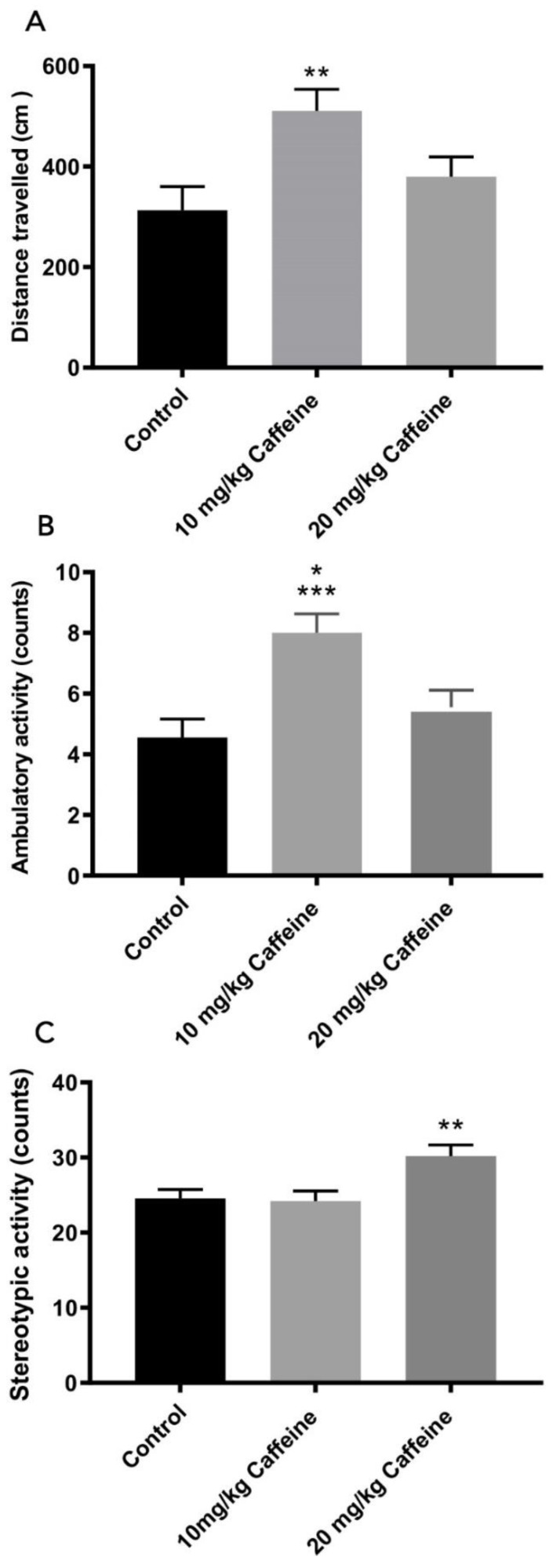
The different components of locomotor activity in control (*n* = 9), 10 mg/kg (*n* = 10) and 20 mg/kg (*n* = 9) caffeine-treated WAG/Rij rats. Data are shown as mean ± SEM. (**A**) ** Significantly longer distance traveled (cm) compared with control rats (*p* < 0.01), (**B**) Higher numbers of ambulatory activity treated with 10 mg/kg caffeine compared with control (*** *p* = 0.001) and 20 mg/kg caffeine-treated WAG/Rij (* *p* < 0.05) rats, (**C**) Higher numbers of stereotypic activity treated with 20 mg/kg caffeine compared with control 10 mg/kg caffeine treatment group and control group (** *p* = 0.01); (one-way ANOVA with post hoc Tukey’s test).

**Figure 4 brainsci-12-00361-f004:**
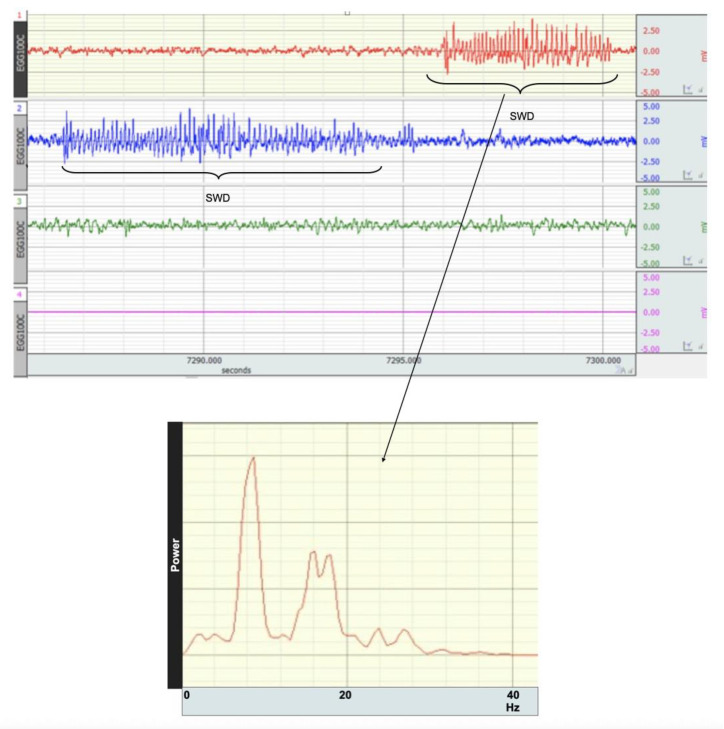
Figure shows examples of SWDs and the power spectra of EEG during the SWD in WAG/Rij rats.

**Figure 5 brainsci-12-00361-f005:**
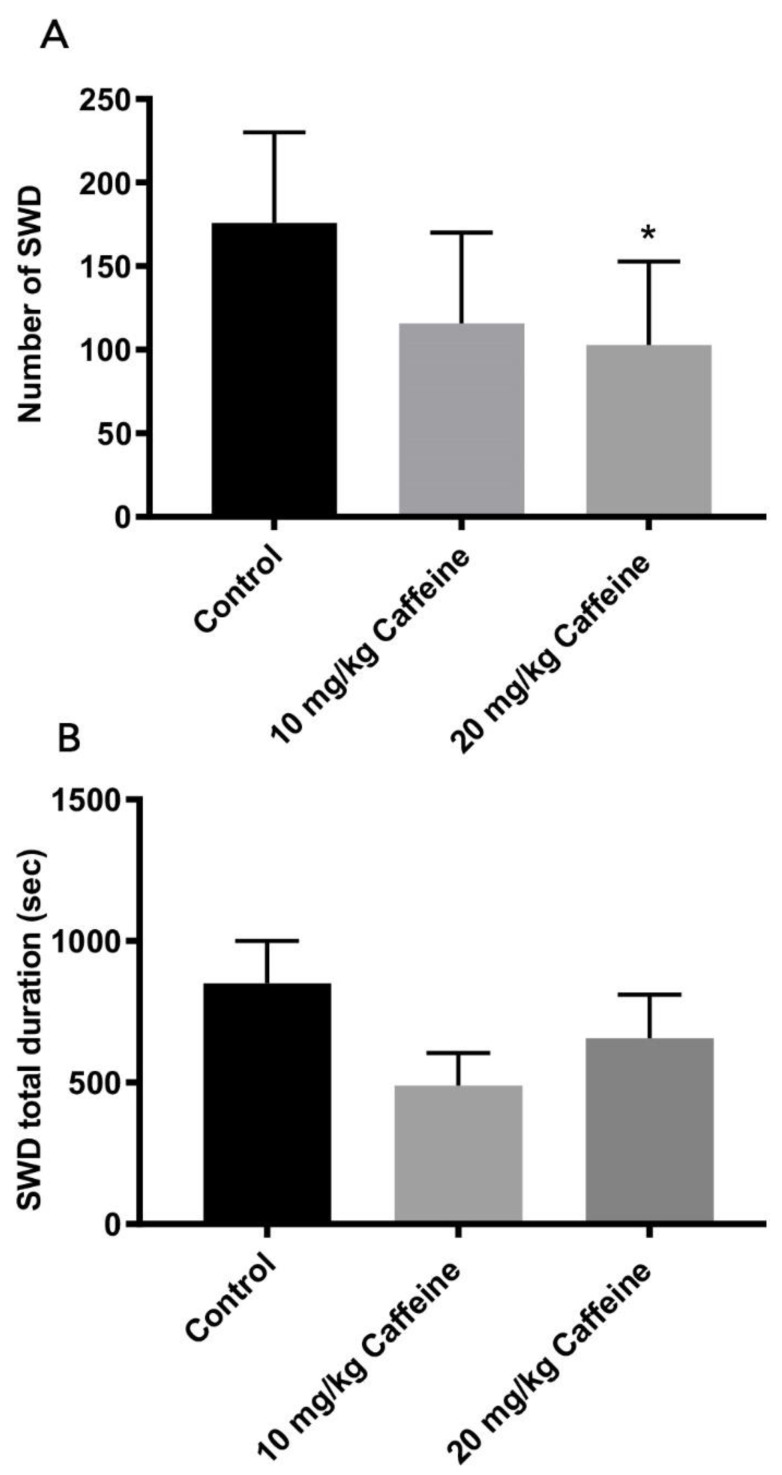
Number (**A**) and total duration (**B**) of SWDs in control (*n* = 7) and 10 mg/kg (*n* = 7) and 20 mg/kg (*n* = 7) caffeine-treated WAG/Rij rats. * *p* < 0.05 in comparison with control group; (one-way ANOVA with post hoc Tukey’s test). Data are means ± S.E.M.

**Table 1 brainsci-12-00361-t001:** Immobility durations of the rats according to the study groups.

Control Group Seconds	10 mg/kg Caffeine Group Seconds	20 mg/kg Caffeine Group Seconds
278	214	168
268	208	207
212	215	200
264	29	207
201	217	156
179	192	156
218	189	130
169	160	157
224	217	169
	139	

## Data Availability

Data are available on request from the corresponding author.

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
