# Peer review of "Effects of Postnatal Caffeine Exposure on Absence Epilepsy and Comorbid Depression: Results of a Study in WAG/Rij Rats"

_brainsci, 2022, doi:10.3390/brainsci12030361_

Round 1

Reviewer 1 Report

See attached file

Author Response

Response to Comments of Reviewer 1:

  • Point 1:

The introduction is not appropriate because does not give clear information to understand the aim of the experiments. At the beginning of the introduction, the authors describe depression as if it is the focus on which the study is based. Indeed depression has been identified as the most frequent psychiatric comorbidity in patients with epilepsy. The aim of this paper is to investigate effect of early caffeine exposure on epileptogenesis and the occurrence of absence seizures and comorbid depression-like behavior in adulthood, therefore it is very important the description of epileptogenic process that is missing. Also it is well documented that despite treatment with anticonvulsant medications aim at different pharmacological targets, approximately one-third of patients remain treatment-resistant; these medications are symptomatic treatments. There is currently no available antiepileptic medications that can prevent, stop, or ameliorate the course of epilepsies (epileptogenic processes) and associated comorbidities. Thus one of the most important benchmarks for epilepsy research agreed upon is the therapy to prevent the development of epilepsy, or anti-epileptogenesis. These important data are missing, thus it is not possible to well understand the purpose of this paper. The authors describe the positive effects of intensive early maternal care on the development of absence epilepsy and depression symptoms in the WAG/Rij pups but they did not report the antiepileptogenic effects of different medications in this animal model (such as antiepileptic medications, antidepressant drugs, anti-inflammatory drugs ecc.) in the background. Please report all these informations.

Introduction is revised according to suggestions of the Reviewer.

  • Point 2

The authors reports in Materials and Methods: “The injections started at 115 PND7 were continued for 5 days. We chose the period of 7-11 days for the intervention since adenosine receptors are present but not fully mature in the various brain areas at this developmental stage [13]”. The reference 13 is not correct because it does not correspond to what is reported in the text. Please insert an appropriate reference.

Reference is changed to `Tchekalarova, J.; Kubová, H. ; Mareš, P. Effects of caffeine on cortical epileptic after discharges in adult rats are modulated by postnatal treatment. Acta Neurol Belg 2013, 113, 493–500`

  • Point 3

How the two doses of caffeine (10 mg/kg and 20 mg/kg) were chosen?

The doses are chosen based on the literature that shows “long-lasting specific changes in brain excitability” with these doses of caffeine.

  • Point 4

In the discussion the authors describe that the effect of caffeine on the development of seizure activity is due to inhibition on adenosine receptors and permanent changes of adenosine receptor subtypes in different brain areas without to well discuss the results and to explain the adenosine receptors effects on neurotransmitters involved in absence epilepsy. It is known that adenosinergic modulatory system influences differently inhibitory GABAergic system and excitatory glutamatergic system in the neocortex and thalamus (important areas in absence epilepsy) of immature brain. The A2AR-induced enhancement of glutamate release from WAG/Rij brain cortex could also play a role in the occurrence of absence epilepsy. It has been reported that thalamocortical excitation is regulated by presynaptic A1 receptors. Also caffeine blocking the A2A receptors, could be to decrease the glutamate release in thalamocortical networks. The expression/activity of A2A receptors is decreased in the somatosensory cortex (focus) and thalamus in presymptomatic WAG/Rij rats. Please discuss this point. The authors write that “seems like early-life caffeine exposure has influence on mechanisms of seizure generation rather than seizure termination” . What are these mechanisms affected by early caffeine treatment?

The discussion section is revised as per suggestions of the Reviewer.

Reviewer 2 Report

    The study is aimed at the investigation of the role of adenosine receptors in genetic epilepsy and comorbid depression in WAG/Rij rats. The authors studied the effects of the adenosine receptor antagonist caffeine administered during the second postnatal week in two doses (10 and 20 mg/kg) on the frequency of spontaneous recurrent seizures and depression-like behaviors in later life. They demonstrate that the caffeine administration in rat pups of WAG/Rij line reduces seizures and depression-like behaviours in adult age.
    The study is overall well-designed, and the results are clearly presented. My main concerns are the following:
    • The type of post-hoc test used must be specified in the section 2.5 of the methods.
    • Line 179 – “p=0.00” is not a valid representation of the confidence level.
    • When the sample size is small, it is preferable to show all values in the figures, not just the mean and the standard error.
    • The number of SWDs and the immobility time and latency in FST are significantly changed only in the 20 mg/kg caffeine group, whereas the sucrose preference was affected only by the 10 mg/kg caffeine administration. This raises the question whether the behavioral changes are due to the reduction in seizures, and not the result of caffeine administration by itself.
    • The limitations of the study must be discussed. For example, the FST, while widely used in antidepressant screening, shows limited correlation with the other behavioral tests for depression-like behavior. In this study, the FST and sucrose preference results are inconsistent with each other when it comes to the dose-dependence, which suggests that one of these tests may not accurately measure the depression-like behavior.

Author Response

Response to Comments of Reviewer 2:

  • Point 1The type of post-hoc test used must be specified in the section 2.5 of the methods.

We used Tukey’s multicomparison test for the post-hoc analysis and added it into section 2.5 as suggested

  • Point 2
    Line 179 – “p=0.00” is not a valid representation of the confidence level.

The p value in row 179 is corrected but do you want us to add mean difference and confidence interval also?

  • Point 3
    When the sample size is small, it is preferable to show all values in the figures, not just the mean and the standard error.

All the values are presented in a table as suggested by the Reviewer

  • Point 4
    The number of SWDs and the immobility time and latency in FST are significantly changed only in the 20 mg/kg caffeine group, whereas the sucrose preference was affected only by the 10 mg/kg caffeine administration. This raises the question whether the behavioral changes are due to the reduction in seizures, and not the result of caffeine administration by itself.

We added an explanation to the discussion section.

  • Point 5
    The limitations of the study must be discussed. For example, the FST, while widely used in antidepressant screening, shows limited correlation with the other behavioral tests for depression-like behavior. In this study, the FST and sucrose preference results are inconsistent with each other when it comes to the dose-dependence, which suggests that one of these tests may not accurately measure the depression-like behavior.

We added requested information to the discussion section.

Round 2

Reviewer 1 Report

the paper may be subject to publication.